# Development and Validation of a Novel Waist Girth-Based Equation to Estimate Fat Mass in Young Colombian Elite Athletes (F20_CA_ Equation): A STROSA-Based Study

**DOI:** 10.3390/nu14194059

**Published:** 2022-09-29

**Authors:** Diego A. Bonilla, Leidy T. Duque-Zuluaga, Laura P. Muñoz-Urrego, Katherine Franco-Hoyos, Alejandra Agudelo-Martínez, Maximiliano Kammerer-López, Jorge L. Petro, Richard B. Kreider

**Affiliations:** 1Research Division, Dynamical Business & Science Society—DBSS International SAS, Bogotá 110311, Colombia; 2Grupo de Investigación NUTRAL, Facultad Ciencias de la Nutrición y los Alimentos, Universidad CES, Medellín 050021, Colombia; 3Research Group in Physical Activity, Sports and Health Sciences (GICAFS), Universidad de Córdoba, Monteria 230002, Colombia; 4Sport Genomics Research Group, Department of Genetics, Physical Anthropology and Animal Physiology, Faculty of Science and Technology, University of the Basque Country (UPV/EHU), 48940 Leioa, Spain; 5Exercise & Sport Nutrition Laboratory, Human Clinical Research Facility, Texas A&M University, College Station, TX 77843, USA

**Keywords:** waist circumference, body composition, DXA scan, kinanthropometry, regression equations, Bayesian analysis, youth sports, athletic performance

## Abstract

The estimation of body fat percentage (%BF) from anthropometry-related data requires population-specific equations to avoid incorrect interpretations in young athletes. Waist girth (WG) has been described as potential predictor of fat mass (FM) in several populations; however, there are no valid WG-based equations to estimate body composition in young Colombian athletes. The aim of this STandardisierte BerichtsROutine für Sekundärdaten Analysen STROSA-based study was twofold: i) to validate the relative fat mass (RFM) and its pediatric version (RFMp) compared to dual-energy x-ray absorptiometry (DXA) and ii) to develop a new equation (F20_CA_) to estimate the fat mass in Colombian children and adolescent elite athletes. A total of 114 young athletes that belong to the ‘Team Medellín’ program (58F, 56M; 51 children, 63 adolescents; 14.85 [2.38] years; 55.09 [12.16] kg; 162.38 [11.53] cm) participated in this cross-sectional study. The statistical analysis revealed a poor correlation, agreement and concordance of RFMp and RFM estimations with DXA measurements. After model specification using both Ordinary Least Square method and Bayesian analysis, the regression output revealed that sex, body mass-to-waist ratio, and waist-to-stature ratio were the statistically significant predictor variables that account for variability in FM. The new F20_CA_ equation is expressed as FM (kg) = 5.46 ∗ (Sex) + 0.21 ∗ (BM/W [kg/m]) + 81.7 ∗ (W/Stature [cm/cm]) − 41.8 (R^2^ = 0.683; SEE = 2.468 kg), where sex is 0 for males and 1 for females. A moderate-to-high correlation and agreement of the F20_CA_ was confirmed within the internal validation data set (R^2^ = 0.689; ICC [95%CI] = 0.805 [0.615, 0.904]; RMSE = 2.613 kg). The Bland–Altman analysis corroborated the high concordance between the reference method (DXA) and the F20_CA_-estimated FM (bias [95% LoA] = 1.02 [−3.77, 5.81] kg), indicating the two methods could be considered interchangeable. Even though external validation is needed, practitioners are advised to use the F20_CA_ in young Colombian athletes with similar characteristics to those who participated in this study.

## 1. Introduction

Childhood and adolescence are a period of human life that ranges from 8 to 19 years age [1]. As part of the physical growth and normal development, child and adolescent populations should become more physically active over time; unfortunately, a worldwide phenomenon is that these populations are insufficiently active [2,3]. In the case of those engaged in physical exercise who initiate training at school age (>8 years), the development of fundamental motor skills might increase the training age of youth while reducing the risk of sport-related injuries [4,5]. Children and adolescent athletes with projection to high performance need strict and constant supervision, not only to ensure their proper growth and development but also to meet their sporting goals [6]. To this end, scientific research has recently focused on providing data regarding adaptive physiological responses to training stimuli [7,8]. Based on those findings, it is of vital importance that multi-disciplinary clinical supervision (athletic coaches, sport nutritionists, sports medicine physicians, etc.) ensures the correct assessment of nutritional status and body composition. This will help to accurately program exercise training or plan dietary interventions based on the individual needs and requirements of young athletes. 

Notably, a recently published systematic review showed that the main factors associated with eating disorders in adolescents are psychological-related variables with a high prevalence of body image dissatisfaction [9]. In fact, in adolescent elite athletes, a higher prevalence rate for eating pathologies in female athletes, in athletes in high-risk sports, and in those aged between 15 to 18 years has been reported [10]. It seems that young athletes at greater risk of eating disorders are those pressured to achieve a body composition that optimizes performance; therefore, nutrition–dietetics practitioners should target education regarding the risk factors of eating disorders, nutritional recommendations, psychological well-being, and the relationship to physical performance [11]. 

Since the early detection of eating disorders is important, the primary care physicians, sports nutritionists and coaches need to be educated and trained to use valid and reproducible techniques in order to avoid the frequently pivotal identification of body composition in adolescent athletes [12]. Kinanthropometry is a discipline focused on performing body measurements and analyze their relationships with the other parameters of health, growth or human movement. It is considered a simple technique that also help to estimate body composition (e.g., body fat percentage [%BF]), monitor maturity status or profile morphological features using absolute data or regression equations [13]. In fact, the International Standards for Anthropometric Assessment has been established by the International Society for the Advancement of Kinanthropometry (ISAK) to reduce the coefficient of variation and improve reliability [14]. Among the different anthropometric measures, it is worth noting the high potential of waist girth (WG) for predicting excess fat mass in humans [15]. WG is considered a fast, simple and inexpensive tool that is directly related to visceral fat [16] and is highly associated with health risk indicators [17]. Although this measurement cannot provide information on body composition, the absolute values of WG have shown a strong correlation (r = 0.80) with the %BF obtained by dual-energy x-ray absorptiometry (DXA) in children and adolescents. Interestingly, regression equations that use the WG as a predictive variable appear to estimate body fat percentage more accurately [18,19,20]. In this regard, Woolcott and Bergman [21] developed and validated a simple equation named relative fat mass (RFM) to estimate the %BF in adults and adolescents between 15–19 years old (RFM = 64 − (20 × (stature [m]/WG [m]))  +  (12 × sex [0 for male and 1 for female])). One year later, the same authors developed a specific equation for children between 8–14 years old (74 − (22 × Stature/WG)  +  (5 × sex)) [22]. Importantly, so far no anthropometry-based equations for estimating body composition in Colombian child and adolescent athletes have been developed [23]. The few research studies in this population focused on the external validation of skinfold-based equations [24,25] or the generation of anthropometric profiles [26,27,28]. Only one study performed the concurrent validation of five equations (none including WG) to estimate the %BF in young Colombian athletes [25]. 

Considering there are no studies that have validated or developed WG-based equations for estimating body composition in children and adolescents, the aim of this study was twofold: (i) to perform the external validation of the RFMp and RFM compared with DXA and (ii) to develop and validate for the first time a new simple WG-based equation to estimate fat mass in young Colombian elite athletes.

## 2. Methods

### 2.1. Study Design

A cross-sectional study was conducted on a single-point measurement of body composition (DXA and anthropometry) in young Colombian athletes with a projection to high performance sport that belong to the ‘Team Medellín’ program (Medellín, Antioquia), in which several of the authors of this article participated. To perform this secondary analysis of data, we followed the analytical workflow of the ‘F20 Project’ (registered in ClinicalTrials.gov under ID #NCT05450588), which is an on-going multi-centric research project that aims to test the validity of current WG-based equations and to develop new models to estimate body composition in populations with different levels of physical activity from various Euro-American countries [29]. We followed the standardized guidelines for reporting secondary data analysis STandardisierte BerichtsROutine für Sekundärdaten Analysen (STROSA) [30], an extension of the Strengthening the Reporting of Observational Studies in Epidemiology (STROBE) statement [31].

### 2.2. Setting

Body composition data was collected from the participants that fulfilled the selection criteria of the ‘Team Medellín’ program between June 2018 and April 2019. ‘Team Medellín’ is a program created in 2017 to monitor children and adolescents with athletic projection in different sports modalities. The analytical procedures for developing and the cross-validation of new equations were conducted as in previous studies carried out by our research group [32,33] and presented under the framework of the ‘F20 Project’ [29]. This study research was conducted as part of the thesis activities of the Master of Science in Sports Nutrition at Universidad CES.

### 2.3. Legal Basis and Data Protection

‘Team Medellín’ is a program led by Medellin’s Town Hall, the Institute of Sports and Recreation INDER Medellín, and the Center for Advanced Studies in Nutrition and Food CESNUTRAL of the Faculty of Nutrition and Food Sciences at Universidad CES. This support program for athletes with a projection for high competitive levels was announced in accordance with Resolutions No 288 of 9 November 2017, No 0622 of 22 November 2017, and the informative circular 001 of 27 November 2017. More information about the program is available at https://www.inder.gov.co/es/team-medellin (accessed on 17 August 2022). Institute of Sports and Recreation INDER Medellín protects and safeguards the personal data in compliance with Colombian Law 1581 of 2012 and its regulatory decrees in connection with Colombian Law 1712 of 2014. The guidance above covers the application of data protection law when using secondary data; however, fully anonymized data was used for research purposes. Since an Institutional Review Board review may be required for a research study that relies exclusively on the secondary use of anonymous information, this study was approved by the Institutional Review Board at the University CES (Act 139 Project: AE-374). 

### 2.4. Selection Criteria and Study Participants

The study population was young male and female Colombian elite athletes. In order to comply with the objectives of this secondary analysis research, the following criteria were taken into consideration for inclusion: (i) residing in Medellin; (ii) belonging to the ‘Team Medellín’; (iii) informed consent signed by both the athletes and their parents or legal representatives; and (iv) being aged between 8–19 years old when measured. The exclusion criteria were: (i) para-athletes; (ii) lack of quality in the records; and (iii) athletes who were evaluated by only one body composition method (DXA or anthropometry). 

### 2.5. Variables

As reference criterion, whole-body 2-compartment body composition was estimated using DXA. The following anthropometric variables established by ISAK were measured: body mass (kg), stature (cm), and WG (cm). Besides the aforementioned, the following anthropometric indexes were included in the regression models as potential predictors: body mass-to-waist (BM/W) and waist-to-stature (W/Stature).

### 2.6. Data Measurement

All the measurements on the selected participants were performed at the Center for Advanced Studies in Nutrition and Food CESNUTRAL at CES University in Medellín, Colombia. All the athletes were evaluated in a similar way at controlled environmental conditions (<24 °C and <60% humidity), with a protocol established for such purpose and the selection criteria were strictly verified. The equipment used was duly calibrated and the procedures were developed in accordance with the latest version of the Declaration of Helsinki [34].

#### 2.6.1. Dual Energy X-ray Absorptiometry

DXA scans were performed following all current technical recommendations and the laboratory procedures reported in previous articles published by our research group [35,36,37]. A Lunar Prodigy™ unit was used (General Electric Healthcare, Madison, WI, USA). Each subject was scanned by a certified bone densitometry technologist (M.K.-L.), and the distinguished bone and soft tissue, edge detection, and regional demarcations were performed using computer software. Test–retest reliability of the DXA unit had a coefficient of variation ranging from 1.0 to 2.0%.

Similar to previous reports [35], we adjusted the DXA measurements on fat-free mass based on the model proposed by Heymsfield et al. [38] to eliminate the influence of fat-free adipose tissue (FFAT). This has been shown to provide more accurate values to detect changes in body composition and for the normalization of physiological variables (e.g., VO_2peak_) in adults [39] and adolescents [40]. In brief, adipose tissue was estimated as DXA-fat mass ÷ 0.85 and then FFAT was calculated as adipose tissue × 0.15. Finally, fat-free mass was adjusted with the subtraction of FFAT. 

#### 2.6.2. Anthropometry

Anthropometric measurements were performed in accordance with the International Standards for Anthropometric Assessment published by the ISAK [14]. The body mass was measured with a digital scale to the nearest 100 g (Seca 874, Hamburg, Germany). A portable stadiometer with a 1 mm graduation was used to measure stature (Seca 213, Hamburg, Germany). WG was measured at the narrowest point between the lower costal (10th rib) border and the top of the iliac crest at the end of a normal expiration with a non-extensible metal tape (Lufkin w606PM, Apex Tool Group, Sparks, MD, USA). The average of two measurements of anthropometric data was calculated and analyzed. The relative technical error of measurement by the ISAK L2 certified anthropometrists was less than 1.5% for measurements [13].

### 2.7. Study Size

Guidelines regarding the sample size needed for accurate predictions have indicated that a total of 130 participants are necessary to obtain a coefficient of determination (R^2^) of 0.5 with an excellent level of prediction using three independent variables [41]. To perform this study, a convenience sample (non-probability sampling) with all 140 young athletes that belonged to the ‘Team Medellín’ was used. However, 26 child and adolescent athletes did not fulfill the inclusion criteria and were excluded. For the external validation of the existing WG-based equations, we divided the remaining set of data (*n* = 114) into two groups that corresponded to each age range of the equations. Therefore, we validated the RFM for adolescents (≥15 years) and RFMp for children (≤14 years) in *n* = 63 and *n* = 51, respectively. To develop the new WG-based equation for Colombian population, the included participants (*n* = 114) were randomly assigned to either the equation development group (EDG, *n* = 83 [75%]) or the validation group (VG, *n* = 31 [25%]). 

### 2.8. Statistical Methods

The descriptive statistics was expressed as mean and standard deviation (SD) unless otherwise is indicated. We used the Yuen–Dixon test [42] with 20% trimmed means (μ_t_) and 20% winsorized standard deviations (σ_w_) as a robust statistical method to compare unequal-sized samples (i.e., EDG [*n* = 83] versus VG [*n* = 31]). This robust statistics provide broader control of Type I error when variances are not equal [43]. 

For the external validation of RFM and RFMp equations, correlation and agreement analyses were performed between the actual and estimated %BF by calculating the correlation coefficient (CC, as Pearson’s r), the coefficient of determination (R^2^), the adjusted coefficient of determination (aR^2^), the standard error of the estimate (SEE), the root mean squared error (RMSE), the intraclass correlation coefficient (ICC) and the concordance correlation coefficient (ρ_c_) as compared to DXA measurements. Bland–Altman diagrams were used for the concordance analysis. This analysis determines whether two measurement methods X and Y agree sufficiently to be declared interchangeable (D = X − Y). The mean of these differences represents the systematic error (bias), while the variance of these differences (1.96 SD) measures the dispersion of the random error.

The following was the set of candidate predictor variables to develop the new model using the Ordinary Least Squares method: age, sex, WG, body mass, stature, BM/W and W/Stature. All possible combinations involving one regressor, two regressors, three regressors and so on until the seven variables were tested. Zellner–Siow prior distributions on the regression coefficients were used to compare all Bayes factors to the null model and all possible models were sorted by their probability from best to worst. For the latter, the Bayesian Adaptive Sampling (BAS) R package was used. The variation explained by the model was determined by the aR^2^. The SEE was calculated for all generated models to measure the regression’s precision, while the RMSE was used to evaluate how estimated values from each equation were close to the actual measured values by DXA. Moreover, all possible regression models were ranked using the akaike information criterion (AIC), the Bayesian information criterion (BIC), the Mallows’ C_p_ and the Hocking’s Sp. After compliance with all of the assumptions of the multiple regression analysis (the normality of residual errors was confirmed with the Omnibus k-squared and Jarque–Bera tests), the model with the best performance was selected for further analysis. The predictability of the selected model was tested in the validation sample by calculating the CC, R^2^, aR^2^, RMSE, CCC and ICC with its respective 95% CI. The concordance analysis was performed using the Bland–Altman diagrams, reporting the concordance intervals at 95% (limits of agreement, LoA). Statistical tests were carried out using the latest version of the environment for statistical computing R [44].

## 3. Results

### 3.1. Participants

A total of 114 child and adolescent Colombian athletes with a projection to a high competitive level fulfilled the inclusion criteria. The sample of the athletes consisted of underwater hockey (*n* = 12), karate (*n* = 12), BMX (*n* = 10), gymnastics (*n* = 10), Taekwondo (*n* = 9), tennis (*n* = 8), ultimate (*n* = 7), swimming (*n* = 6), diving (*n* = 5), table tennis (*n* = 5), archery (*n* = 4), chess (*n* = 4), fencing (*n* = 3), wrestling (*n* = 3), badminton (*n* = 3), bowling (*n* = 3), squash (*n* = 3), judo (*n* = 2), athletics (*n* = 1), bike trial (*n* = 1), BMX freestyle (*n* = 1), speed skating (*n* = 1) and agility (*n* = 1) players. All athletes were enrolled in high-performance competitive tournaments at the national or international level. Table 1 shows the characteristics of all EDG and VG participants. 

No statistically significant differences were found between EDG and VG participants. An exploratory correlation analysis was performed for a quick and simple summary to understand the relationship between the study variables in all participants (Table 2). BM/W showed the high significant correlation values with other variables (r ≥ 0.72).

### 3.2. External Validation of the RFMp and RFM for Colombian Children and Adolescents

The RFM equation for adults has been reported to be useful for estimating %BF in adolescents from 15 to 19 years of age. Complementarily, the RFMp has been developed as a modified version of the original equation for children and adolescents between 8 and 14 years of age. However, our statistical analysis showed very low and low correlation for RFMp (Figure 1A) and RFM (Figure 1B), respectively, in young Colombian athletes. In fact, the ICC and CCC revealed a very poor and poor agreement and concordance with DXA measurements for RFMp (ICC [95% CI]: 0.148 [−0.131, 0.406]; CCC [95% CI]: 0.145 [0.030, 0.313]) and RFM (ICC [95% CI]: 0.536 [0.333, 0.691]; CCC [95% CI]: 0.531 [0.370, 0.662]), respectively. The Bland–Altman analyses showed a too wide range of the 95% CI for the LoA for RFMp (−15.922, 14.111) and RFM (−9.644, 13.093). 

### 3.3. New Waist Girth-Based Equation to Estimate Fat Mass in Young Athletes

After the evaluation of all models with all possible combinations of predictor variables to estimate fat mass in kilograms (127 models in total), the best performance was obtained in the model that included Sex, BM/W and W/Stature as explainable variables for estimating fat mass. Table 3 shows the performance metrics of the top ten models with three regressors to estimate fat mass (see all possible models including all regressors in Appendix A). All models are expressed as Ŷ = β_0_ + β_1_X_1_ + β_2_X_2_ + β_3_X_3_. It is important to note that models with maximum three regressors were prioritized for simplicity and considering the a priori sample size calculation. Even though we also evaluated all possible combinations to estimate %BF or fat-free mass as independent variables (data not shown), the selected model to estimate fat mass outperformed all other possibilities after the model specification process.

A complementary Bayesian approach was implemented to select the most suitable regression model. Zellner–Siow prior distributions on the regression coefficients were used to compare all Bayes factors to the null model and all possible models were sorted by their probability from best to worst (Figure 2). This alternative analysis revealed that the best model for estimating fat mass was the one that included sex, the BM-to-W and W-to-Stature ratios.

Once the predictor variables were selected, we ensured the assumptions of the linear regression were satisfied. Normality of residual errors was confirmed with the Omnibus K-squared (*p* = 0.586) and Jarque–Bera (*p* = 0.579) tests which are based on skewness and kurtosis, respectively. VIFs revealed there were no multicollinearity issues. Finally, autocorrelation in the residuals was discarded using the Durbin–Watson test where values between 1.5 and 2.5 are considered as acceptable. Finally, the heteroscedasticity condition was rejected (*p* = 0.287). The statistical parameters of the regression are shown in Table 4. The new model (F20_CA_) to estimate FM in Colombian children and adolescent athletes is shown in the following Equation (1), where 0 for men and 1 for women (SEE = 2.468 kg):FM (kg) = 5.46 ∗ (Sex) + 0.21 ∗ (BM/W [kg/m]) + 81.7 ∗ (W/Stature [cm/cm]) − 41.8(1)

### 3.4. Validation of the F20_CA_ Equation

The validation process of the F20_CA_ equation resulted in a moderate-to-high level of correlation and concordance with DXA measurements. Regarding goodness of fit, the new equation had low values of SEE (2.471 kg) and RMSE (2.613 kg) with moderate-to-high aR^2^ (0.679), which indicated the model fit for the estimation. In addition, the C.b was equal to 0.963 (very close to 1). The C.b is a bias correction factor that measures how far the best-fit line deviates from a line at 45 degrees. Furthermore, there was a good agreement and correlation between measured and estimated fat mass based on the obtained ICC (95% CI) of 0.805 (0.615, 0.895). Finally, the Bland–Altman analysis corroborated the high concordance between the new equation and DXA (bias = 1.02 kg), since 100% of the data remain within a relatively short range of the 95% LoA (−3.771, 5.815 kg) (Figure 3).

## 4. Discussion

Young athletes with projection to a high competitive level require early monitoring to understand individual responses to exercise programs, morphological changes [27,28] and the development of technical skills [46,47]. Therefore, the assessment of nutritional status and body composition is extremely important to avoid unwarranted nutrient deficiencies, chronic low energy availability (LEA) and eating disorders [48]. Sports practitioners frequently rely on equations to estimate body composition (i.e., fat mass, musculoskeletal mass) as an accessible, practical and non-expensive methodology with moderate correlation to reference methods [23]; however, it is clear that the estimation of %BF without a population-specific equation or correction factor may lead to incorrect interpretations [49]. It should be emphasized that using wrong equations to assess body composition may affect the dietary planning process by resulting in inaccurate athlete resting energy expenditure or macronutrient distributions that may further evoke in chronic LEA or macro/micronutrients deficiencies [50]. 

Considering the aforementioned situations, this study aimed to validate and develop WG-based equations in young Colombian elite athletes. Although RFM/RFMp has been associated with biochemical-related and cardiometabolic parameters in children with chronic kidney disease [51] or with overweight/obesity [52], the findings of our external validation process revealed that neither RFM nor RFMp are recommended for young Colombian athletes given the poor correlation, agreement and concordance with the DXA scans (see Figure 1). Similar to our results, recent studies performed in south American population have also concluded that the RFM equation seems not to be valid for estimating %BF in adolescents. In a total of 631 individuals (197F, 434M) aged 11 to 18, Ripka et al. [49] evaluated the validity and accuracy of different equations to estimate %BF from anthropometric data against DXA. Their findings revealed that RFM underestimated %BF in boys and girls and had lower sensitivity compared to BMI and tri-ponderal mass index. In another study, Encarnação et al. [53] evaluated 420 Brazilian adolescents (204F, 216M) aged 15–19 years to verify the validity of anthropometric methods, including the RFM, with DXA as reference method. The authors concluded that RFM was not valid for predicting %BF in the studied sample regardless of sex or age. It worth noting that RFM has been reported as a tool to diagnose high adiposity [54], which might explain the poor estimation performance in children and adolescents due to their lower adiposity. Nevertheless, further research is warranted to make definitive conclusions in this regard.

Considering the origin and number of studies, a recent scoping review by Cerqueira et al. (2022) showed that only a few South American countries have developed anthropometric equations to estimate FM, %BF or body density in young populations: Argentina (1), Uruguay (1), Chile (2) and Brazil (9). No studies have been performed in Colombia to validate or develop regression equations based on anthropometric data to evaluate body composition in children and adolescents. Thus, the F20_CA_ is the first equation for athletic Colombian children and adolescents.

In the young Colombian athletes who participated in this study, as age increases, fat mass increases in women (r = 0.6), while in men, a positive correlation was observed between fat-free mass and age (r = 0.787). Although chronological age has been found to explain to a large extent the changes in body composition [55], our statistical analysis (traditional and Bayesian) showed that age was not a statistically significant variable that could be used as a regressor. In fact, the Bayesian approach revealed that sex, the BM/W and W/Stature ratios were the variables with higher marginal posterior inclusion probabilities, indicating that these are important for explaining the data variance or prediction (estimation). Both ratios have been found to have a strong correlation with body composition as well as strong predictors of abdominal obesity across the lifespan [56,57,58,59,60,61,62]. Thus, we were able to successfully develop and validate a novel equation (F20_CA_) to estimate fat mass in kilograms with relatively low SEE (2.471 kg) and RMSE (2.613 kg), which indicate that the estimated values are close to the DXA measurements. Our concordance analysis suggests that both fat mass measurement methods (DXA and the F20_CA_-estimated values) can be interchanged. 

### 4.1. Limitations and Strengths

The results of this study should be discussed in light of the following limitations and strengths. Firstly, we are aware that as a result of the skewed selection of participants this study is susceptible to bias and other forms of selection errors; however, the sample had relatively homogeneity between number of male and female participants. In addition, sample was representative for young Colombian elite athletes. Secondly, even though we might have experienced attrition bias given we did not reach the target sample size (*n* = 130), we obtained a higher R^2^ than initially planned, as well as moderate-to-high values of correlation, agreement and concordance between the developed equation and DXA measurements. Moreover, the general rule of thumb is ≈50 participants for developing a regression equation [63]. A larger sample size for external validation is warranted for generalizability in the young athletic Colombian population, particularly including young athletes from different geographical locations. It is necessary to point out the need for validation current or new WG-based equations in other populations (e.g., health and disease). Thus, following the methodological procedures of the ‘F20 Project’ might increase generalizability and scientific soundness at the time of evaluating body composition in populations with different levels of physical activity by validating and/or developing new WG-based models [29]. Third, the study was performed as secondary data analysis, although we were able to answer the specific research question/objective. 

The development and validation of new models, such as the F20_CA_ equation, increase the accuracy in the estimations, the scientific soundness in research projects and generalizability to evaluate the young Colombian athletic population. Aware of the need for external validation, we consider that the simple developed/validated F20_CA_ equation could become a formally established tool by the Institute of Sports and Recreation INDER Medellín and the Colombian Ministry of Sports.

### 4.2. Practical Recommendations

The following are practical take-home points for sport and performance practitioners and scientists who supervise young athletes without access to more sensitive, accurate and expensive methods:Follow the International Standards for Anthropometric Assessment (ISAK protocol). The International Olympic Committee research working group on body composition, health and performance recommends the procedures established by the ISAK [64]. However, it is necessary to point out that WG measured at different sites (minimal [ISAK], midway, iliac, umbilicus) does not affect the relationships with visceral adipose tissue measured by magnetic resonance imaging and with cardiometabolic risk factors in children and adolescents, regardless of race or sex [65].Estimate fat mass with the F20_CA_ as: FM (kg) = 5.46 ∗ (Sex) + 0.21 ∗ (BM/W [kg/m]) + 81.7 ∗ (W/Stature [cm/cm]) − 41.8, where sex is zero for men and one for women.Use a fat mass index (fat mass/stature^2^, kg/m^2^) chart analysis to avoid the misclassification of young players’ nutritional status in order to assess differences between age categories and to design individual intervention for fat loss or muscle gain [66,67].Use the sum of skinfolds to accurately detect changes in body composition (i.e., %BF) during a nutrition and/or exercise intervention. The sum of skinfolds has been shown to strongly correlate with %BF measured by DXA in well-trained athletes [68,69,70].Estimate the young athlete’s fat-free mass (kg) according to the expression: Body Mass [kg] − F20_CA_-estimated fat mass [kg].Use the estimated fat-free mass to calculate the energy (e.g., resting energy expenditure or energy expenditure during exercise) [71,72,73] and macronutrient distribution [74,75]. For the selection of any predictive equation to estimate body composition or energy expenditure, the practitioner should consider the following: (a) the assessed athletes should be similar to the ones used to develop the original equation; (b) similarities in age and sex; (c) similarity in adiposity and physical activity levels; (d) the technique or protocol used in the study; and (e) the equipment used [70].Use the estimated fat-free mass to calculate the energy availability to avoid chronic LEA and the subsequent relative energy deficiency in sport. The energy availability is calculated as: Energy Intake − Exercise Energy Expenditure/Fat-Free Mass [76,77].Calculate the skinfold-corrected muscle girths (girth − [π × skinfold/10]) to monitor changes in musculoskeletal mass [78].Body composition is important but is not the cornerstone of sports success. Track other psychological, physiological and performance-related variables to gain a more integrative picture of the athlete’s adaptive response to exercise, nutrition and rest (allostatic response) [79,80].

## 5. Conclusions

Current scientific evidence on the estimation of %BF encourages researchers and practitioners to advocate for the creation of simple and specific equations to children and adolescents [23]. We successfully formulated a new simple, specific and WG-based two-component model to estimate body composition in Colombian children and adolescent elite athletes using ordinary least squares regression. The regression output revealed that sex, the BM/W ratio and the W/Stature ratio were the statistically significant predictor variables that account for variability in the fat mass of young athletes. The new equation was named as F20_CA_ and showed a moderate-to-high correlation and agreement within the internal validation data set (R^2^ = 0.689; ICC [95%CI] = 0.805 [0.615, 0.904]; RMSE = 2.613 kg). Furthermore, all data measurements were within the concordance limits between values measured with the reference method (DXA) and the F20_CA_-estimated fat mass (bias [95% LoA] = 1.02 [−3.77, 5.81] kg), indicating that the two methods could be considered interchangeable. While the equation is externally validated in other geographical population groups, practitioners are advised to use the F20_CA_ in Colombian children and adolescents with similar characteristics to those who participated in this study.

## Figures and Tables

**Figure 1 nutrients-14-04059-f001:**
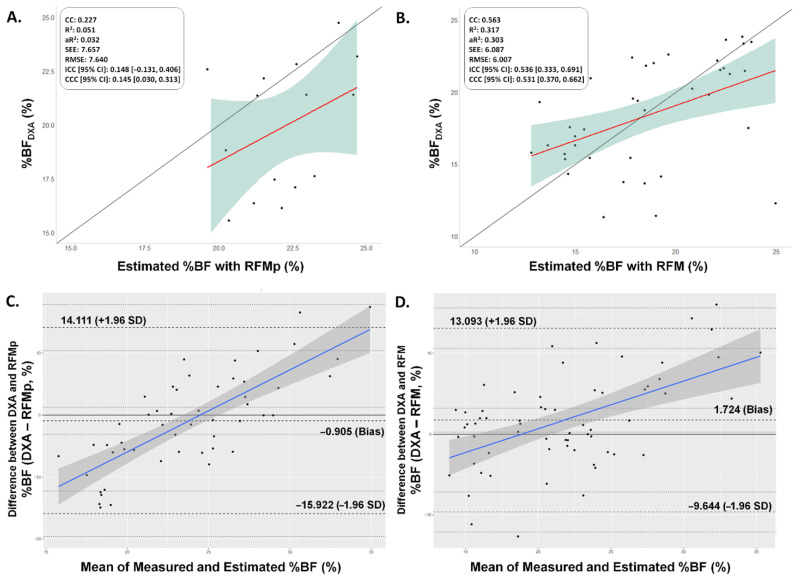
External validation of the RFMp and RFM in young Colombian athletes. (**A**) Concordance correlation plot of DXA measurements and estimations with RFMp. (**B**) Concordance correlation plot of DXA measurements and estimations with RFM. The solid 45° line represents perfect concordance while the red line represents the linear regression line through the observations. CC: Correlation coefficient; R^2^: Coefficient of determination; R^2^a: Adjusted coefficient of determination; SEE: Standard error of the estimation; RMSE: Root mean square error; ICC: Intraclass correlation coefficient; CCC: Concordance correlation coefficient (ρ_c_). (**C**) Bland–Altman plot for differences between measured and estimated %BF with the values obtained by the RFMp. (**D**) Bland–Altman plot for differences between measured and estimated %BF with the values obtained by the RFM. Individual differences between real and estimated %BF values are plotted against the mean of the values of measured and estimated %BF. Limits of Agreement are shown as dashed black lines with 95% confidence intervals, bias (as dashed black line) with 95% confidence interval, and regression fit of the differences on the means (as solid blue line).

**Figure 2 nutrients-14-04059-f002:**
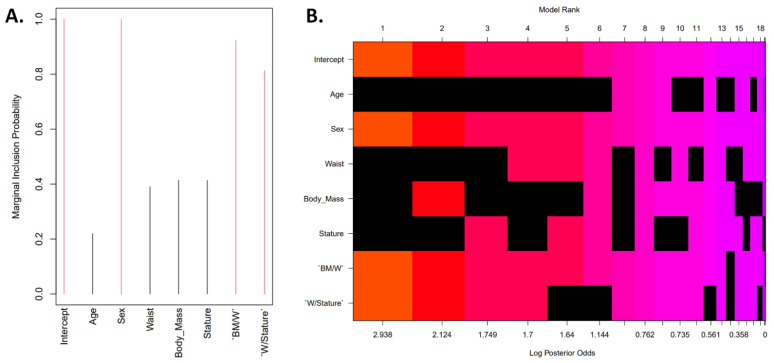
Visualization of the model space with all of the potential predictors. (**A**) Marginal posterior inclusion probabilities for each of the covariates. The marginal posterior inclusion probabilities greater than 0.5 are shown in red. (**B**) Zellner–Siow prior distributions on the regression coefficients. The rows correspond to each of the variables and intercept (labels on the y-axis) while the x-axis corresponds to the possible models. The models are sorted by their posterior probability from best (left) to worst (right) with the rank on the top x-axis (each column represents one model). Excluded variables in a model are shown in black for each column and the variables included are colored (the color is related to the log posterior probability with orange as the highest probability model). BM: body mass; BM/W: body mass-to-waist ratio; W: waist girth; W/Stature: waist-to-stature ratio.

**Figure 3 nutrients-14-04059-f003:**
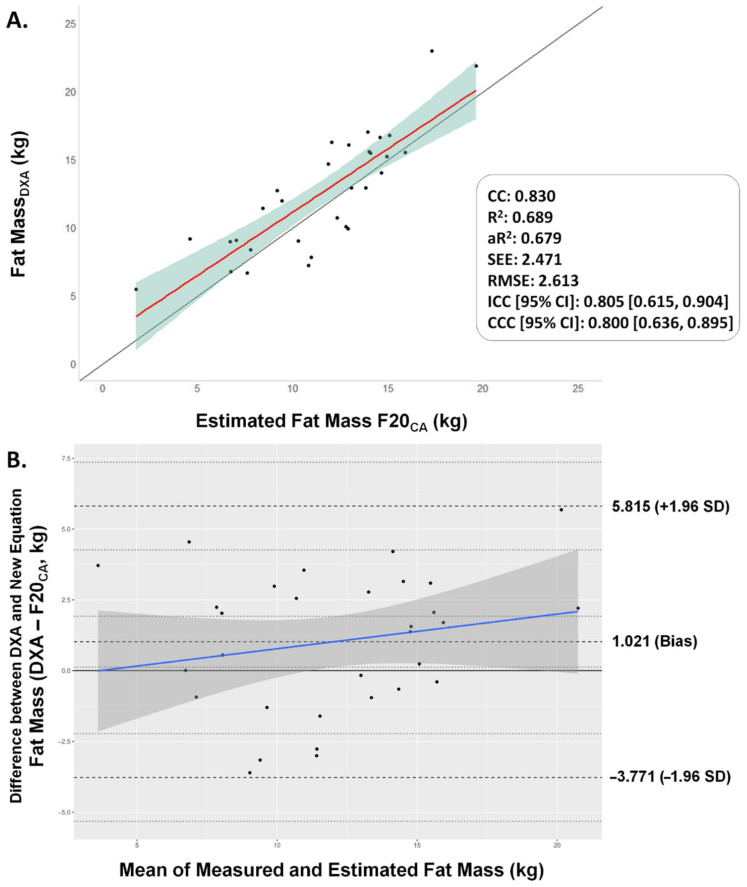
Correlation and concordance analysis of the F20_CA_ equation. (**A**) Concordance correlation plot. CC: Correlation coefficient; R^2^: Coefficient of determination; R^2^a: Adjusted coefficient of determination; SEE: Standard error of the estimation; RMSE: Root mean square error; ICC: Intraclass correlation coefficient; CCC: Concordance correlation coefficient (ρ_c_). (**B**) Bland–Altman plot for differences between measured and estimated fat mass in kilograms with the developed equation (F20_CA_). Individual differences between real and estimated fat mass values are plotted against the mean of the values of measured and estimated fat mass.

**Table 1 nutrients-14-04059-t001:** Characteristics of the study population.

Variable	All (*n* = 114) $\overset{¯}{X}$ (SD) [95% CI]	EDG (*n* = 83) $\overset{¯}{X}$ (SD) [95% CI]	VG (*n* = 31) $\overset{¯}{X}$ (SD) [95% CI]	ES_t_ (MoE_Δ_) [95% CI]	*p* Value
*Sex*					
Women	58 (50.87%)	46 (55.42%)	10 (32.25%)		
Men	56 (49.12%)	37 (44.57%)	21 (67.74%)		
*Race*					
White-Mestizo	110 (96.49%)	79 (95.18%)	31 (100.0%)		
Afro-descendant	4 (3.50%)	4 (4.81%)	0 (0.0%)		
*CA*					
Children (8 to 14)	51 (44.73%)	39 (46.98%)	12 (38.70%)		
Adolescents (15 to 19)	63 (55.26%)	44 (53.01%)	19 (61.29%)		
Age	14.85 (2.38)	14.87 (2.30)	14.79 (2.60)	0.10 (1.15) [−1.05, 1.26]	0.850
Body mass	55.09 (12.16)	54.51 (12.02)	56.63 (12.60)	2.97 (4.49) [−1.52, 7.47]	0.189
Stature	162.38 (11.53)	161.80 (11.42)	163.91 (11.89)	4.02 (4.34) [−0.31, 8.37]	0.069
Waist	69.04 (6.30)	68.77 (6.46)	69.74 (5.88)	1.29 (2.57) [−1.28, 3.86]	0.314
BM/W (m/m)	79.02 (11.88)	78.52 (11.59)	80.37 (12.73)	3.11 (4.81) [−1.69, 7.93]	0.197
W/Stature (cm/cm)	0.42 (0.02)	0.42 (0.02)	0.42 (0.02)	0.00 (0.01) [−0.008, 0.01]	0.666
FM (kg)	12.39 (4.34)	12.33 (4.35)	12.54 (4.36)	0.47 (2.06) [−1.58, 2.54]	0.639
%BF (%)	22.65 (6.40)	22.80 (6.53)	22.26 (6.11)	−0.55 (3.33) [−3.88, 2.78]	0.738

Data is presented as mean ($\overset{¯}{X}$) and standard deviation (SD) unless otherwise is indicated. The effect size (ES_t_) corresponds to the difference between the two trimmed means (μ_t2_ − μ_t1_) in original units. %BF: percentage of body fat; BM/W: body mass-to-waist ratio; CA: children and adolescents; CI: confidence interval; EDG: equation development group; FM: fat mass; MoE_Δ_: marge of error for the CI on the difference between the two trimmed means; VG: validation group; W/Stature: waist-to-stature ratio. Statistically significance (*p* < 0.05 of the two-tailed *p* value) for difference between EDG and VG.

**Table 2 nutrients-14-04059-t002:** Means, standard deviations and correlations with confidence intervals.

Variable	$\overset{¯}{X}$	SD	1	2	3	4	5	6
1. Age	14.85	2.38						
2. Body mass	55.09	12.16	0.70 *					
			[0.59, 0.78]					
3. Stature	162.38	11.54	0.61 *	0.88 *				
			[0.49, 0.72]	[0.82, 0.91]				
4. Waist	69.04	6.30	0.56 *	0.89 *	0.72 *			
			[0.42, 0.67]	[0.85, 0.93]	[0.61, 0.79]			
5. BM/W	79.02	11.89	0.72 *	0.95 *	0.89 *	0.72 *		
			[0.62, 0.80]	[0.93, 0.97]	[0.84, 0.92]	[0.62, 0.80]		
6. W/Stature	0.43	0.03	0.10	0.28 *	−0.11	0.62 *	0.02	
			[−0.09, 0.28]	[0.11, 0.45]	[−0.29, 0.08]	[0.49, 0.72]	[−0.16, 0.21]	
7. FM_DXA	12.39	4.34	0.41 *	0.50 *	0.27 *	0.44 *	0.50 *	0.32 *
			[0.24, 0.55]	[0.35, 0.63]	[0.09, 0.43]	[0.28, 0.58]	[0.35, 0.63]	[0.14, 0.47]

Data is presented as mean ($\overset{¯}{X}$) and standard deviation (SD). Values in square brackets indicate the 95% confidence interval for each correlation. The confidence interval is a plausible range of population correlations that could have caused the sample correlation [45]. BM/W: body mass-to-waist ratio; FM: fat mass; W/Stature: waist-to-stature ratio. * Statistically significance (*p* < 0.01).

**Table 3 nutrients-14-04059-t003:** Regression results to estimate fat mass using DXA as the criterion.

OLS Equation	R^2^	aR^2^	C_p_	AIC	BIC	hsp
Ŷ = β_0_ + β_1_(Sex) + β_2_(BM/W) + β_3_(W/Stature)	0.683	0.671	4.045	393.525	405.619	0.080
Ŷ = β_0_ + β_1_(Sex) + β_2_(BM) + β_3_(Stature)	0.661	0.648	9.439	399.004	411.098	0.085
Ŷ = β_0_ + β_1_(Sex) + β_2_(BM) + β_3_(W/Stature)	0.655	0.642	10.950	400.476	412.570	0.087
Ŷ = β_0_ + β_1_(Sex) + β_2_(Stature) + β_3_(W/Stature)	0.587	0.572	27.862	415.385	427.479	0.104
Ŷ = β_0_ + β_1_(Sex) + β_2_(W) + β_3_(BM/W)	0.585	0.570	28.304	415.741	427.835	0.104
Ŷ = β_0_ + β_1_(Sex) + β_2_(W) + β_3_(W/Stature)	0.585	0.569	28.367	415.791	427.885	0.104
Ŷ = β_0_ + β_1_(Sex) + β_2_(W) + β_3_(Stature)	0.584	0.568	28.671	416.034	428.129	0.105
Ŷ = β_0_ + β_1_(Sex) + β_2_(W) + β_3_(BM)	0.582	0.566	29.078	416.360	428.454	0.105
Ŷ = β_0_ + β_1_(Age) + β_2_(Sex) + β_3_(W)	0.574	0.558	31.182	418.021	430.115	0.110
Ŷ = β_0_ + β_1_(Sex) + β_2_(BM) + β_3_(BM/W)	0.563	0.546	33.914	420.129	432.224	0.110

Data was generated using the *ols_step_all_possible* function of the ‘olsrr’ v0.5.3 R package. It tested all possible subsets of the set of potential independent variables. AIC: akaike information criterion; aR2: adjusted coefficient of determination; BIC: bayesian information criterion; BM: body mass; C_p_: Mallows’ C_p_; hsp: Hocking’s Sp; OLS: Ordinary Least Squares; R^2^: coefficient of determination; W: waist girth.

**Table 4 nutrients-14-04059-t004:** Regression results to estimate fat mass using DXA as the criterion.

Predictor	*b* [95% CI]	*beta* [95% CI]	*sr^2^* [95% CI]	*r*	*R^2^*	a*R^2^*	VIFs	DW
(Intercept)	−41.80 [−51.69, −31.91] *				0.683 [0.55, 0.75] *	0.671		1.877
Sex	5.46 [4.29, 6.63] *	0.63 [0.49, 0.76]	0.35 [0.20, 0.49]	0.38 *			1.132	
BM/W	0.21 [0.16, 0.26] *	0.55 [0.43, 0.68]	0.30 [0.16, 0.44]	0.48 *			1.026	
W/Stature	81.70 [60.91, 102.49] *	0.52 [0.39, 0.65]	0.25 [0.12, 0.37]	0.35 *			1.105	

A significant *b*-weight indicates the beta-weight and semi-partial correlation are also significant. *aR*^2^: adjusted coefficient of determination; *b*: represents unstandardized regression weights; *beta*: indicates the standardized regression weights; BM/W: body mass-to-waist ratio; DW: Durbin–Watson; *r*: represents the zero-order correlation; *sr^2^*: represents the semi-partial correlation squared; VIFs: variance inflation factors; W/Stature: waist-to-stature ratio. * Statistically significance (*p* < 0.01).

## Data Availability

The data that support the findings of this study are available upon request from the corresponding author.

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
