# Peer review of "Development and Validation of a Novel Waist Girth-Based Equation to Estimate Fat Mass in Young Colombian Elite Athletes (F20CA Equation): A STROSA-Based Study"

_nutrients, 2022, doi:10.3390/nu14194059_

Round 1

Reviewer 1 Report

Regarding the manuscript, I make the following comments:

 1.          The study discussed an interesting topic, a current problem.

2.           The relationship between abdominal circumference and cardiac risk is a well known experience, it raises the possibility of estimating the whole fat mass from trunk dimension, the relationship between WG and body fat.

 3.          In my opinion, the sample studied is not suitable for discussing the topic, because these are elite, young athletes who, including girls, show a body fat value of 22%. Experience has shown that the sample studied is slightly larger than average in terms of age.

 4.          The significant statistical apparatus was also unable to prove the predictive power of the abdominal circumference, certainly the relationship between body dimensions and circumference played a role (see the correlation matrix).

 5.           In principle, the authors could have had the opportunity to isolate the degree of visceral fat based on DEXA measurements, which, by the way, is an obvious factor in obesity.

 6.          Even though the methodology of data analysis is up-to-date, the limitation is very strong. Don't interpret the authors' view that validity can only be expected from similar individuals. It is a hypotesis.

Author Response

Dear reviewer, 

Thanks for your comments.

Regarding the manuscript, I make the following comments:

  1. The study discussed an interesting topic, a current problem.

Response: We appreciate the reviewer’s comments. Appropriate changes have been made to address the reviewer’s feedback on a point-by-point basis, with relevant changes highlighted in the revised manuscript.

  1. The relationship between abdominal circumference and cardiac risk is a well known experience, it raises the possibility of estimating the whole fat mass from trunk dimension, the relationship between WG and body fat.

Response: You raise an important point here. However, it should be noted that this research is not focused on cardiovascular risk or include overweight/obese population.

  1. In my opinion, the sample studied is not suitable for discussing the topic, because these are elite, young athletes who, including girls, show a body fat value of 22%. Experience has shown that the sample studied is slightly larger than average in terms of age.

Response: Thanks for your comments. We disagree with the reviewer since this study aimed to validate current WG-based equations (RFM and RFMp) and develop a new model for estimating body composition in Colombian young elite athletes. The importance of body composition assessment in this athletic population is clearly stated in the introduction section:

Children and adolescent athletes with projection to high performance need a strict and constant supervision not only to ensure their proper growth and development but also to meet their sporting goals (Logan et al. 2019)

  • Logan, K.; Cuff, S.; Council On Sports, M.; Fitness. Organized Sports for Children, Preadolescents, and Adolescents. Pediatrics 2019, doi:10.1542/peds.2019-0997.

As the reviewer should be aware, obesity and the risk of cardiovascular disease are beyond the scope of this study.

  1. The significant statistical apparatus was also unable to prove the predictive power of the abdominal circumference, certainly the relationship between body dimensions and circumference played a role (see the correlation matrix).

Response: Thanks for the comment. We partially agree with the reviewer’s observations. As an exploratory analysis, correlation only evaluated the linear relationship between two variables. Therefore, this is not the way to “prove” the predictive power of any variable. To do so, both frequentist and Bayesian approaches were used to identify the statistically significant regressors (see Table 3 and Figure 2). For example, the Bayesian-based objective evaluation of the most suitable regression model (predictive variables) is stated in the paper:

“Zellner-Siow prior distributions on the regression coefficients were used to compare all Bayes factors to the null model and all possible models were sorted by their probability from best to worst”.

In agreement with the reviewer, waist girth alone was not superior to other ratios that included this variable (i.e., body mass-to-waist [BM/W] and waist-to-stature [W/Stature]). In any case, our aim was fulfilled given we were able to develop a new simple waist girth-based equation by including BM/W and W/Stature. 

  1. In principle, the authors could have had the opportunity to isolate the degree of visceral fat based on DEXA measurements, which, by the way, is an obvious factor in obesity.

Response: Thanks for the comment. Even though we are aware of the well-described associations between visceral adipose tissue and health risk (quantified by RMN or CT as more sensitive and valid techniques than DXA), we still do not understand the suitability of the comment. Again, it should be noted that this research is not focused on the assessment of cardiovascular risk, obesity or the evaluation of overweight/obese population. Thus, these topics are beyond the scope of this study.

  1. Even though the methodology of data analysis is up-to-date, the limitation is very strong. Don't interpret the authors' view that validity can only be expected from similar individuals. It is a hypotesis.

Response: Thanks for the comments. We would appreciate if the reviewer expands his/her insights given none limitation (beyond the confusion with obesity or cardiovascular risk assessment) is mentioned in the report.

On the other hand, “validity can only be expected from similar individuals” is something that does not correspond to a hypothesis. As recommended by several authors (Marfell-Jones et al. 2012; Salamat et al. 2015; Cerqueira et al. 2022), including previous research performed by our Research Division (Bonilla et al. 2022a; Bonilla et al. 2022b), population-specific equations are needed to avoid incorrect interpretations when evaluation children and adolescents as well as any other given population. Therefore, estimations can only be considered as valid if a similar population to the original one used to develop the regression equation is used. Accurately, Kobel et al. (2022) emphasized:

Researchers and practitioners should be cautious of using population-specific equations to estimate body fat at the individual level. The validity of such equations relies on several assumptions…”.

  • Marfell-Jones, M., Nevill, A. M., & Stewart, A. D. (2012). Anthropometric surrogates for fatness and health. In Body composition in sport, exercise and health (pp. 146-166). Routledge.
  • Salamat, M. R., Shanei, A., Salamat, A. H., Khoshhali, M., & Asgari, M. (2015). Anthropometric predictive equations for estimating body composition. Advanced biomedical research, 4.
  • Bonilla, D. A., De León, L. G., Alexander-Cortez, P., Odriozola-Martínez, A., Herrera-Amante, C. A., Vargas-Molina, S., & Petro, J. L. (2022a). Simple anthropometry-based calculations to monitor body composition in athletes: Scoping review and reference values. Nutrition and Health, 28(1), 95-109.
  • Cerqueira, M. S., Amorim, P. R., Encarnação, I. G., Rezende, L. M., Almeida, P. H., Silva, A. M., ... & Marins, J. C. (2022). Equations based on anthropometric measurements for adipose tissue, body fat, or body density prediction in children and adolescents: a scoping review. Eating and Weight Disorders-Studies on Anorexia, Bulimia and Obesity, 1-18.
  • Bonilla, D. A., Duque-Zuluaga, L. T., Muñoz-Urrego, L. P., Moreno, Y., Vélez-Gutiérrez, J. M., Franco-Hoyos, K., ... & Petro, J. L. (2022b). Development and Validation of Waist Girth-Based Equations to Evaluate Body Composition in Colombian Adults: Rationale and STROBE–Nut-Based Protocol of the F20 Project. International Journal of Environmental Research and Public Health, 19(17), 10690.
  • Kobel, S., Kirsten, J., & Kelso, A. (2022). Anthropometry–assessment of body composition. Deutsche Zeitschrift für Sportmedizin/German Journal of Sports Medicine, 73(3), 106-111.

Reviewer 2 Report

Waist girth (WG) was used as a potential predictor of fat mass (FM) in several populations, but there  are no valid WG-based equations to estimate body composition in young Colombian athletes. This study is very important, estimations were done precisely and understandable.

From my point of view, this paper is ready to be published, after improving small language mistakes and text editing errors. 

Author Response

Response to Reviewer 2 Comments

Waist girth (WG) was used as a potential predictor of fat mass (FM) in several populations, but there are no valid WG-based equations to estimate body composition in young Colombian athletes. This study is very important, estimations were done precisely and understandable.

From my point of view, this paper is ready to be published, after improving small language mistakes and text editing errors.

Response: Appreciate the reviewer’s comments. We are glad to know that the reviewer considers this article suitable for publication. A native speaker has revised the manuscript to improve readability.

Round 2

Reviewer 1 Report

To the authors: Thank you above all for your replies to my comments. I accept all your views. There is no doubt in my mind that their research on the subject is correct and their opinion is moderate. With regard to the limitations, I maintain my opinion that there is no general description of the populations for which the findings should be interpreted. This finding does not apply to the sample examined.   Beszúrás